# Taxation and Enterprise Innovation: Evidence from China's Value-Added Tax Reform

Ke Ding [1] , Helian Xu [1,*] and Rongming Yang [2]

1   School of Economics and Trade, Hunan University, Changsha 410006, China; b12250020@hnu.edu.cn
2   International Student Education and Administration Office, Yunnan University of Finance and Economics, Kunming 650221, China; zz1726@ynufe.edu.cn
*   Correspondence: xuhelian@hnu.edu.cn; Tel.: +86-13037319676

**Abstract:** This article used China as an example to study how tax reform affects the innovative behavior of companies. Our research showed that value-added tax (VAT) reform can affect corporate innovation behavior. On the basis of patent-application data of Chinese enterprises, we used the difference-in-differences framework to study the differences in the performance of Chinese industrial enterprises in patent applications before and after China's 2009 VAT reform. We demonstrated that China's VAT reform had a positive impact on corporate innovation; this conclusion is robust. In subsequent research, we demonstrated that the VAT reform promoted corporate innovation by expanding corporate investment in fixed assets and reducing corporate debt ratios; however, due to the Chinese government's subsidies to corporations and financing constraints, the pecking-order effect of corporate innovation was increased. In addition, the VAT reform had a greater impact on the innovation of export enterprises and non-state-owned enterprises. This research provided insights for emerging countries into formulating innovation-driven sustainable development tax reduction policies.

**Keywords:** VAT; reform; enterprise innovation; China; difference-in-differences

## 1. Introduction

The use of the design of fiscal policies to influence corporate behavior is widespread in countries around the world. The governments of some large economies believe that appropriate fiscal policies are conducive to economic development. In particular, as the largest emerging country, the Chinese government began to pay attention to the direct impact of subsidies, taxes, and other fiscal policies on enterprises more than two decades ago, and defined these fiscal policies as mainstays of government policy [1]. Generally speaking, the purpose of the Chinese government in formulating these fiscal policies was to conform to the trend of economic development; however, after China joined the WTO in 2001, more fierce market competition also requires tax reforms that are more in line with the market economy. Therefore, a series of taxation reforms began in the early 21st century. Among these tax reforms, that with the most obvious direct impact on enterprises is the value-added tax (VAT) reform.

The value-added tax is a turnover tax levied on the value-added amount generated using the circulation of goods (including taxable services) as the tax basis. Since China's basic tax system reform in 1994, the value-added tax has been the most important tax in China's tax system. Moreover, before the 2004 VAT pilot reform, China's value-added tax was used nationwide, following the system of levying production value-added tax on enterprises established during the 1994 tax reform. The establishment of the VAT system in 1994 was to curb the economic risk of overinvestment in China at that time. However, with the development of China's economy, this tax system that inhibits corporate investment seems a bit untimely. Therefore, after the pilot implementation in some regions in 2004 and 2007, in 2009, the Chinese government implemented a VAT reform nationwide to

eliminate the problems caused by the productive VAT system adopted by the Chinese government in the 1994 tax reform. The productive VAT inhibited Chinese companies' investment in equipment upgrades and reduced the ability of Chinese companies to upgrade technology and expand production [2]. To solve these problems, the Chinese government proposed, for the first time, to allow enterprises the ability to deduct the VAT contained in newly purchased equipment. Additionally, it canceled the VAT exemption for imported equipment and the VAT refund policy for foreign-invested enterprises purchasing domestic equipment and uniformly reduced the VAT collection rate to 3% for small-scale taxpayers.

Regarding the impact of this VAT reform, most of the literature has focused on its investment impact on enterprises and the impact of enterprises on exports [3–5]; however, few researchers have investigated its impact on corporate innovation. Thus, this paper used the combined database of the Annual Survey of Industrial Firms (ASIF) and the database of China Innovative Enterprise to comprehensively study the effect and mechanism of the 2009 VAT reform on enterprise innovation by adopting the method of the difference-in-differences (DID) model.

This paper used the difference in the impact of the 2009 VAT reform on enterprises with different ownership to select the treatment group and the control group of the DID model. As a result of the VAT reform in 2009, the VAT exemption policy for imported equipment and the VAT refund policy for foreign-invested enterprises purchasing domestic equipment were canceled. The changes in these policies show that domestic companies are more affected by the 2009 VAT reform than foreign-invested companies. Therefore, we distinguished between domestic companies and foreign companies as the treatment group and control group, respectively, of the DID model. The reason for choosing the aforementioned DID model is that we were able to eliminate other influencing factors in the model and obtain additional information regarding innovation performance of domestic enterprises. The additional innovation performance can be attributed to the VAT reform.

We used a direct measurement method to define enterprise innovation performance; specifically, the number of enterprise patent applications was used to measure enterprise innovation performance [6]. By analyzing empirical evidence, we found the following. First, the VAT reform in 2009 did promote the innovation performance of Chinese domestic companies. Additionally, the dynamic analysis demonstrated that the treatment group companies and the control group companies had the same development trend before the 2009 VAT reform; after the 2009 VAT reform, however, the innovation performance of the treatment group companies and the control group companies showed different development trends. The number of patent applications of companies increased significantly compared with the control group companies two years after the 2009 VAT reform. Second, by reviewing the research on the impact mechanism, we found that the VAT reform promoted the innovation performance of the treatment group companies by expanding the company's fixed-asset investment and reducing the company's debt ratio. Third, by analyzing the impact of VAT reform on the quality of corporate innovation, we found that the promotion of the innovation performance of Chinese enterprises by the VAT reform was mainly concentrated in areas of low-quality innovation, such as appearance patents and new utility patents. For invention patents with relatively high innovation quality, the promotion effect due to the VAT reform was lower. In this research, we confirmed that this difference in innovation was due to the government subsidy that had an undesirable pecking-order effect on corporate innovation. Fourth, the VAT reform had a more significant promotion effect on exporters and non-state-owned enterprises (SOEs), but the impact on non-exporters and SOEs was not significant.

The main contributions of this article are as follows. (1) This article enriches the relevant literature on the impact of tax incentives on corporate innovation. The literature on this issue has mainly focused on the impact of corporate R&D tax credits on corporate innovation. Moreover, most of the studies have confirmed that tax incentives represented by R&D tax credits have a positive impact on corporate innovation [7–10]. However, these

studies do not discuss the impact of VAT reform on enterprise R&D. The research in this article supplements the literature. (2) The research in this article expands the research on the impact of China's VAT reform on enterprises. In recent years, the literature on the effects of a series of VAT reforms in China has been increasing. Most of the studies have focused on the impact of VAT reforms on corporate investment and corporate exports [3–5]. The research in this article focuses on the impact of China's VAT reform on corporate innovation. (3) The research in this article also supplements the research on corporate innovation. This paper finds that the innovation of enterprises under financing constraints has a pecking order effect. Notably, corporate innovation is similar to export, and both are high-input and high-risk investment projects. The recent trade literature has demonstrated that, when companies with financing constraints export, they rank their export targets and prefer to export to the market with the greatest profit [11,12]. Learning from these studies, we posit that when companies choose innovation strategies, there may also be a pecking-order effect on innovative investment choices, that is, choosing innovation strategies in line with the maximization of corporate interests. The empirical conclusions of this article also confirm our conjecture: the existence of the pecking-order effect on this innovative investment choice, which is a new supplement to enterprise innovation research.

The remainder of this article is structured as follows. The second section is on the policy background and theoretical hypotheses. After reviewing the history of China's VAT reform and the literature on existing tax policies affecting corporate innovation, we proposed three theoretical hypotheses. The third section presents the empirical strategy, which mainly introduces the research methods of empirical research. The fourth section presents the main empirical research results and robustness test of this article and the fifth section presents a further discussion. We discuss the pecking order effect of corporate innovation and the heterogeneous impact of VAT reform on corporate innovation. The sixth section is the conclusion of this article.

## 2. Policy Background and Theoretical Hypotheses

### 2.1. Policy Background: China's VAT Reform in 2009

In China, tax reform is an important macro-control tool. China first introduced the VAT policy in 1979. Until 1994, China's VAT policy was only implemented in some key industries and export industries. The year 1994 was significant for China's VAT reform, as the Chinese government implemented a standardized VAT system nationwide. In this VAT reform, the Chinese government introduced a rare production-type VAT in the production process of enterprises [4]. Although the 1994 VAT reform achieved the Chinese government's goals of preventing macroeconomic overheating and increasing fiscal revenues, it also caused problems, the most obvious of which was that companies were required to pay taxes twice when they invested in fixed assets; one tax was a direct tax that applied when the company purchased assets, and the other was an indirect taxation of goods produced by companies using these assets. This double taxation inhibited corporate investment in fixed assets [5].

With the changes in the Chinese economy from 2004 to 2009, to adapt to the new development situation, the Chinese government gradually introduced new VAT reforms throughout the country. The most obvious feature of this VAT reform is the shift of the VAT from production to consumption, eliminating the problem of repeated taxation. This reform was implemented in the heavy industry of the three northeastern provinces (i.e., Liaoning, Jilin, and Heilongjiang), was expanded to six central provinces in 2007 (i.e., Shanxi, Anhui, Jiangxi, Henan, Hubei, and Hunan), and, in 2009, was implemented throughout the country.

Our research strategy was to treat the 2009 VAT reform as a quasi-natural experiment. In addition, the difference in the policy impact experienced by enterprises with different ownerships in the reform was used to construct experimental treatment groups and control groups. In the value-added reform in 2009, the government canceled some favorable policies for foreign-invested enterprises, including the exemption of value-added import equipment tax and VAT refund policies for foreign-invested enterprises purchasing domestically-made equipment. Therefore, foreign-funded enterprises benefited less

than domestic enterprises in this reform. Based on this difference in policy impact, we treated domestic companies as the treatment group and foreign-invested enterprises as the control group.

*2.2. Theoretical Hypothesis*

Because innovation is the basic driving force of economic growth [13], economists have begun to pay attention to the mechanism of influence on enterprise innovation very early. Regarding the influencing factors of enterprise innovation, the literature has mainly included the following categories. Some scholars emphasized the positive effects of knowledge diffusion and R&D spillovers [14–17]; other scholars emphasized the impact of intra-industry competition on enterprise innovation. This includes the impact of creative destruction and corporate R&D dynamics on corporate innovation [18–23]; in addition, some scholars have been concerned about the negative impact of financing constraints on corporate innovation [24–28].

Based on the aforementioned viewpoints and the possible actual impact of tax reform, this paper posits that China's VAT reform in 2009 may have affected corporate innovation in the following ways:

(1) The VAT reform has reduced the cost of using fixed capital and, therefore, will promote enterprises' investment in fixed capital [4]. This increase in investment in fixed assets also means that the VAT reform will increase the rate at which incumbents update production equipment and increase the amount of capital equipment put into production. According to the research on the impact of knowledge spillovers and R&D spillovers on enterprise innovation, after companies accelerate the rate of updating production equipment and increase the amount of capital equipment invested in production, they can increase R&D capabilities by introducing more intermediate products and capital equipment that reflect foreign knowledge and obtain useful information [15]. This kind of enhancement in R&D capabilities of enterprises due to knowledge spillovers will incentivize enterprises to innovate. In addition, the VAT reform has reduced the cost of fixed assets, expanded the investment level of incumbents, and promoted intensified competition among companies in the same industry. According to the research on the influence of intra-industry competition on enterprise innovation, when the incumbent enterprises within the industry have little technological difference before the impact of the VAT reform, the intensification of inter-firm competition will also incentivize enterprise innovation [6,20].

In summary, the VAT reform will encourage enterprises to innovate by motivating enterprises to invest in fixed assets.

(2) The VAT reform has alleviated the level of corporate debt. This occurs because, on the one hand, the VAT reform has reduced the level of taxes and fees faced by companies, and, on the other hand, the VAT reform has reduced the depreciation rate of corporate fixed assets because of the decline in the partial irreversible rate of assets [5]. These cost reductions for companies will reduce the level of corporate debt and increase the level of corporate cash flow. According to the literature on the impact of cash flow on corporate innovation, increasing the level of corporate cash flow reduces the maintenance cost of corporate innovation activities. Therefore, mitigating corporate debt levels can improve corporate innovation [28,29].

Based on the aforementioned theoretical analysis, we propose two research hypotheses:

**Hypothesis 1.** *China's VAT reform promotes corporate innovation; China's VAT reform promotes corporate innovation by promoting corporate fixed-asset investment and alleviating corporate debt levels.*

(3) Enterprise innovation requires a large amount of investment in innovation; the intensity of enterprise innovation is also related to the intensity of market competition, and the market scale is related to the company's R&D capabilities [6,30,31]. Since the domestic Chinese companies in our treatment group had poorer R&D capabilities than the foreign companies in the control group, the domestic companies will face higher innovation

risks in the face of increasingly competitive markets [32]. In this case, although the VAT reform enabled domestic companies to expand investment and provided incentives to innovate [5], these companies must also consider the benefits of innovation in the presence of innovation risks [33]. Therefore, although the VAT reform has promoted enterprise innovation, due to the existence of these factors, the investment in industry innovation will lag significantly [34]. This lag in innovation investment will make the growth of enterprise innovation results, such as the number of patent applications, also lag behind. From this, we formulate Hypothesis 2:

**Hypothesis 2.** *The VAT reform promotes corporate innovation, but due to the existence of innovation risks, corporate innovation is lagging. This will be reflected in the fact that the number of enterprise patent applications will not increase significantly until many years after the completion of the VAT reform.*

(4) China's VAT reform promotes corporate innovation by reducing the cost of corporate investment in fixed assets and reducing corporate debt and other financing constraints. However, a long-ignored issue is the impact mechanism of tax reform on corporate innovation quality choices.

Some studies have demonstrated that, under the condition of financing constraints, companies prudently choose high-risk investment projects such as innovation and export. For example, an article pointed out that financing constraints made some developing countries' enterprises choose between exports and innovation [35]. Another article pointed out that companies chose export destinations under financing constraints to have a pecking order effect; specifically, companies rank the expected return of export destinations and export to the country with the largest expected return [12].

Drawing on the ideas of the aforementioned articles and considering the similarities between exports and innovation, we posit that both belong to the early stage of high-input and high-risk corporate investment behavior. We believe that, in the case of very severe financing constraints of Chinese companies [36], although companies have alleviated part of their debt levels in the VAT reform, they will still choose the projects with the largest expected benefits for innovation, that is, companies in the VAT reform. The subsequent innovative choices also have a pecking-order effect.

In addition, because the Chinese government often promotes corporate innovation through selective industrial policies such as industrial subsidies, Chinese companies will conduct not only innovative behaviors for the purpose of promoting technological progress and maintaining competitive advantages but also strategic innovative behaviors aimed at obtaining government subsidies [37]. Moreover, the Chinese government's investigation of the innovation capability of enterprises tends to be biased toward the number of enterprise patents. Therefore, enterprises do not pay attention to the quality of innovation and choose lower-quality innovations to help enterprises obtain government subsidies. In this case, the pecking-order effect encourages enterprises to conduct lower-quality innovations. Therefore, based on the aforementioned analysis, we propose the third hypothesis:

**Hypothesis 3.** *There is a pecking-order effect in corporate innovation. Additionally, if a higher number of innovations of enterprises results in them more easily obtaining government subsidies, the VAT reform will incentivize enterprises to conduct low-quality innovations rather than high-quality innovations.*

### 3. Empirical Strategy

*3.1. Data and Variables*

The samples studied in this article were from the database of ASIF and China Innovative Enterprise Database. ASIF is calculated by the National Bureau of Statistics of China and covers most Chinese companies with annual sales of more than 5 million RMB (more than 20 million RMB after 2011). The China Innovative Enterprise Database is recorded by

the China Microeconomic Data Query System of the Easy Professional Superior (EPS) data platform (the website of the data platform is: http://microdata.sozdata.com/login.html accessed on: 14 March 2021). The database covers patent applications and authorization data of most innovative companies. These patent data are collected by the China National Intellectual Property Administration. Through merging and sorting, we obtained a sample of all innovative companies from 2005 to 2013, including company production data and company patent data. Next, we cleaned the samples. To achieve this objective, we excluded samples that did not fulfill the accounting standards; specifically, we deleted any of the following observation samples with negative values: total asset minus liquid asset, total asset minus net fixed asset, accumulated depreciation minus current-year depreciation, and total paid in capital minus paid in capital from each type of investor.

After cleaning the data, we obtained 103,254 samples containing 64,295 independent companies.

The explanatory variable in the empirical study in this paper was the number of enterprise patent applications that had undergone logarithmic processing. The reason we used the logarithmic number of corporate patent applications as the proxy variable of corporate innovation performance is that the number of patent applications applied for by companies as the output of corporate innovation can be used as a direct measure of corporate innovation performance [38].

We counted three types of corporate patents recorded in the China Innovative Enterprise Database. The number of all three types of patent applications represents the overall level of enterprise innovation performance; thus, we used them in the main regression. From the detailed data of all three types of patent applications, we learned from the ideas of Tan et al. [39] and distinguished the quality of enterprise innovation by the technological content of enterprise innovation. Among them, because the invention patent has the highest technical content, we used its value as an indicator to measure the high innovation quality of the enterprise. The other two types of patents, namely new utility patents and design patents, have low technical content; the sum of the two types of patents was used as an indicator to measure the low innovation quality of enterprises.

This paper used the DID model to identify the impact of VAT reform on enterprise innovation. Therefore, based on the aforementioned information, we selected domestic companies as the treatment group of the model and foreign-invested companies as the control group of the model. Based on this classification, we set the dummy variable treat. When the enterprise was a treatment group enterprise, the value was taken as one, and when it was a control group enterprise, the value was taken as zero. We also set a time dummy variable T. When the year was before 2009, the value was taken to be zero, and when the year was after 2009, it was taken to be one. By multiplying the aforementioned two dummy variables, we obtained the key explanatory variable treat*T and used it in this article.

We also used control variables that may have affected corporate innovation: corporate labor productivity (LBR), measured by the ratio of corporate operating income to the number of employees in the company; corporate age (age), determined by the companies measured by the year of the current year minus the opening time; the size of the enterprise, measured by the natural logarithm of the enterprise; the liquidity ratio (lique), measured by the ratio of current assets to current liabilities; and the SA index, constructed based on the ideas of Hadlock and Pierce [40]. We also constructed dummy variables of SOEs, time, province, and three-digit industry. The specific descriptive statistics are shown in Table 1.

**Table 1.** Summary statistics.

| Variable | Definition | Mean | Sd |
|---|---|---|---|
| Patents | | | |
| patent | log (all patents + 1) | 1.325 | 1.094 |
| patent 2 | log (invented patents + 1) | 0.515 | 0.784 |
| patent 1 | log (non-invented patents + 1) | 1.075 | 1.065 |
| Firm's variables | | | |
| size | Log of a firm's total assets | 11.533 | 1.568 |
| age | The age of enterprise | 2.318 | 0.671 |
| lique | Current assets/Current debt | 11.687 | 1527.231 |
| SA | SA Index | −2.776 | 0.459 |
| LBR | Operating income/employee | 6.113 | 1.169 |

*3.2. Typical Facts*

According to the definition of the explained variables in the prior section, we can use the method of drawing to briefly describe the impact of the 2009 VAT reform on the innovation performance of the treatment group and the control group. Figures 1–3 show the trend of changes in the average number of patent applications by companies in the treatment group and the control group during the sample interval from 2005 to 2013. The following typical facts were found. (1) Generally speaking, the average number of all patent applications and non-invention patent applications of the treatment group enterprises (i.e., domestic enterprises) was smaller than that of the control group (i.e., foreign-invested enterprises). However, companies in the treatment group had a slight advantage over the companies in the control group in the average number of invention patent applications. (2) The average number of all patent applications and the average number of non-invention patents had relatively similar time trends. Before 2009, the average number of patent applications by companies in the treatment group was significantly smaller than the average number of patent applications by companies in the control group. However, since 2010, the gap between the two has been significantly reduced, which suggests that the number of patent applications of treatment group companies increased significantly after 2009. (3) The number of all patent applications and non-invention patent applications of the treatment group reached their peak in 2012, which may indicate that the impact of the VAT reform on enterprise innovation may have a delayed effect, similar to the content described in Hypothesis 2. (4) The difference from the above is that the average number of invention patent applications belonging for high-quality innovations did not produce significant differences within the sample interval. In particular, the gap between the average number of invention patent applications by companies in the treatment group and the average number of invention patent applications by companies in the control group were the smallest in 2009 and 2013, while a large gap remained between 2010 and 2012.

The aforementioned typical facts preliminarily suggest that the VAT reform in 2009 improved the innovation performance of the treatment group enterprises, but this improvement in innovation performance was likely to be driven by lower-quality innovative behaviors. To further verify whether this typical fact reflects the real impact of VAT reform on enterprise innovation, we used the empirical model presented in the next section to conduct a rigorous empirical test.

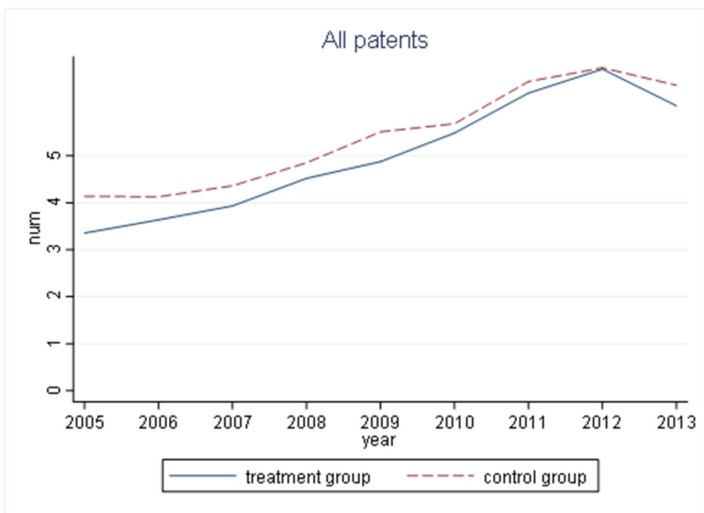

**Figure 1.** Trends of all patents.

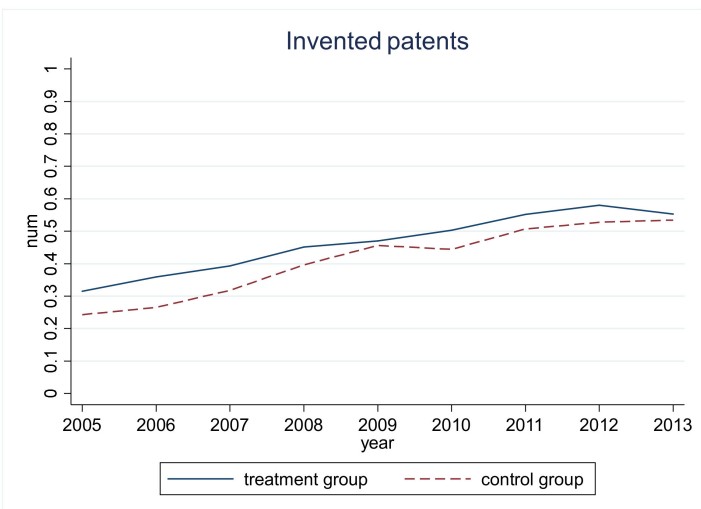

**Figure 2.** Trends of invented patents.

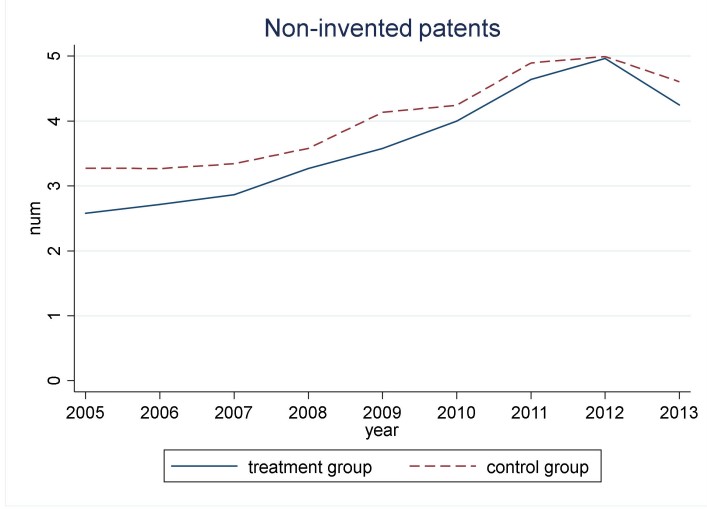

**Figure 3.** Trends of non-invented patents.

### 3.3. Empirical Model

The objective of this paper was to analyze the relationship between tax policy and corporate innovation by studying the impact of China's nationwide value-added tax reform on corporate innovation in 2009. Since China's 2009 VAT reform abolished the free policy of import VAT for foreign-invested enterprises and the preferential policy of VAT rebate for purchases of domestic equipment, these policies rendered the benefits of foreign-invested enterprises in this reform less than that of domestic enterprises. This difference in policy benefits between companies provided a rare opportunity for us to directly study the causal relationship between tax policy and corporate innovation in a DID framework.

On the basis of the aforementioned analysis, we determined the control group and treatment group of the DID model. To further identify the impact of VAT reform on enterprise innovation and prove the theoretical hypotheses, we borrowed ideas from Zhang et al. [4] and Chen et al. [5] and constructed the following DID model as the benchmark model of this article:

$$y_{icjt} = \beta_1(treat * post) + \beta_c Controls + a_{ct} + a_{jt} + \delta_i + \varepsilon_{it} \tag{1}$$

where $y_{icjt}$ is the total number of patent applications of enterprise $i$ in industry $j$ in province $c$ in year $t$, and treat represents an enterprise-level grouping variable. According to the aforementioned grouping, we treated the enterprises in the treatment group, that is, domestic enterprises as 1 and foreign-invested enterprises as 0. Post is the time grouping variable of the policy. We considered 2005–2008 as 0, and 2009–2013 as 1. Controls are control variables. $a_{ct}$ is the province-year fixed effect, and $a_{jt}$ is the 3-digit industry-year fixed effect. We used these two cross-term fixed effects to control for the province and industry trends that change over time and to avoid missing variables. $\delta_i$ is the fixed effect of the individual enterprise. To solve the possible heteroscedasticity problem, we used robust standard errors for clustering. Our main concern was the coefficient of $\beta_1$, which represents the average treatment effect of the VAT reform in 2009 on enterprise innovation.

In addition, considering the possible lag of VAT reform on enterprise innovation and to test whether the DID model satisfies the parallel trend condition, we adopted the event-study method pioneered by Jacobson et al. [41] to study the dynamic impact of VAT reform on enterprise innovation. The model of the event study method can be expressed as:

$$y_{icjt} = \sum_{t=2005}^{t=2013} \beta_t(treat * post) + a_{ct} + a_{jt} + \delta_i + \varepsilon_{it} \tag{2}$$

Consistent with the above, $a_{ct}$ is the province-year fixed effect, $a_{jt}$ is the 3-digit industry-year fixed effect, and $\delta_i$ is the fixed effect of the individual enterprise.

## 4. Empirical Results

### 4.1. Results of the Benchmark Model

Table 2 shows the regression results of the benchmark model. To ensure the accuracy of the results of our benchmark model, we respectively used the Pool-OLS model and fixed effects model to estimate the benchmark model. The first and second columns of Table 1 show the results of the estimation using the pool-OLS model. In the first column of regression, we controlled for the fixed effects of province, industry, and year; in the second column of the model, we controlled for the province-year fixed effects and industry-year fixed effects. Additionally, when using the Pool-OLS model to estimate, we borrowed the ideas of Petersen [42] and clustered at the enterprise level to obtain the standard error. The third and fourth columns of Table 3 show the results of the fixed effects (FE) model. Similar to the first and second columns, we controlled for different fixed effects in the third and fourth columns, respectively. The third column is similar to the first column. We controlled for the fixed effects of province, industry, and year, while the fourth column controlled the

fixed effects of province-year and industry-year. For the fixed effects model, this paper used robust standard errors and clustered at the enterprise level.

**Table 2.** Benchmark model's result.

| Variable | (1) Patent | (2) Patent | (3) Patent | (4) Patent |
|---|---|---|---|---|
| treat * post | 0.158 *** | 0.161 *** | 0.094 *** | 0.092 *** |
| | (13.409) | (13.471) | (2.772) | (3.767) |
| Size | 0.022 ** | 0.021 ** | −0.076 *** | −0.075 ** |
| | (2.207) | (2.101) | (−2.620) | (−2.528) |
| age | −0.129 *** | −0.129 *** | −0.084 *** | −0.088 *** |
| | (−21.759) | (−21.479) | (−2.871) | (−2.994) |
| lique | 0.000 | 0.000 | 0.000 *** | 0.000 *** |
| | (1.026) | (0.756) | (11.125) | (8.554) |
| SA | 0.571 *** | 0.580 *** | 1.148 *** | 1.097 *** |
| | (15.106) | (15.145) | (11.331) | (10.894) |
| LBR | 0.000 | 0.005 | 0.016 * | 0.022 ** |
| | (0.072) | (1.245) | (1.701) | (2.417) |
| Constant | 1.963 *** | 1.730 *** | 5.961 *** | 5.479 *** |
| | (8.313) | (3.914) | (8.277) | (3.799) |
| Year dummy | YES | NO | YES | NO |
| province dummy | YES | NO | YES | NO |
| Industry dummy | YES | NO | YES | NO |
| province * Year | NO | YES | NO | YES |
| industry * Year | NO | YES | NO | YES |
| N | 103,254 | 103,254 | 103,254 | 103,254 |
| Adj.R$^2$ | 0.107 | 0.111 | 0.045 | 0.069 |
| Estimation method | Pool-OLS | Pool-OLS | FE | FE |

Note: *, **, and *** refer to significance at the 1%, 5%, and 10% levels, respectively.

**Table 3.** Robustness test.

| | (1) | (2) | (3) | (4) | (5) |
|---|---|---|---|---|---|
| Variable | patent | patent | patent | patent | patent |
| treat * post | 0.080 ** | 0.090 ** | 0.127 *** | 0.084 ** | |
| | (2.420) | (2.457) | (3.293) | (2.002) | |
| treat * predict | | | | −0.018 | |
| | | | | (−0.411) | |
| treat * post 2008 | | | | | 0.047 |
| | | | | | (1.258) |
| size | 0.009 | −0.052 | −0.089 *** | −0.075 ** | −0.069 ** |
| | (0.333) | (−1.592) | (−2.728) | (−2.539) | (−2.328) |
| age | −0.111 *** | −0.070 ** | −0.063 ** | −0.088 *** | −0.092 *** |
| | (−4.114) | (−2.224) | (−1.994) | (−2.993) | (−3.141) |
| lique | 0.000 *** | 0.000 *** | 0.000 *** | 0.000 *** | 0.000 *** |
| | (8.349) | (8.352) | (8.018) | (8.545) | (8.574) |
| SA | 0.661 *** | 1.054 *** | 1.124 *** | 1.098 *** | 1.088 *** |
| | (6.956) | (9.458) | (10.286) | (10.902) | (10.812) |
| LBR | 0.021 ** | 0.024 ** | 0.024 ** | 0.022 ** | 0.023 ** |
| | (2.366) | (2.313) | (2.297) | (2.420) | (2.433) |
| constant | 2.262 * | 4.453 *** | 6.467 *** | 4.708 *** | 4.675 *** |
| | (1.923) | (3.117) | (8.666) | (3.650) | (3.617) |
| province * Year | YES | YES | YES | YES | YES |
| industry * Year | YES | YES | YES | YES | YES |
| N | 100,406 | 85,056 | 82,353 | 103,254 | 103,254 |
| Adj.R$^2$ | 0.065 | 0.068 | 0.084 | 0.069 | 0.069 |

Note: *, **, and *** refer to significance at the 1%, 5%, and 10% levels, respectively.

In Table 2 demonstrates that, regardless of which estimation method is used, the coefficient of treat * post is significant and positive at the 1% level. This finding shows that China's VAT reform in 2009 significantly improved the innovation performance of domestic enterprises, and the innovation performance of domestic enterprises increased by approximately 0.1.

Since Hypothesis 2 suggests that the impact of tax reform on corporate innovation may have a delayed impact, we must study the dynamic effects of VAT reform on corporate innovation. In addition, because the DID model requires that the treatment group and the control group have the same time trend before the event impact, the parallel trend hypothesis must be established. To verify the aforementioned dynamic effects and parallel trends, we used the aforementioned event-study method proposed to test them.

According to the research ideas of the event-study method, we set the initial year of VAT reform in 2009 as the "year when the policy occurred", then generated the variables for each year, and finally added these year-by-year variables to equation (2) for regression. If (2) the estimated regression coefficients of the variables in each year before 2009 were not significant within the 95% confidence interval, the parallel trend hypothesis of the benchmark model in this paper is valid.

The results of the event-study method are shown in Figure 4 We demonstrated that before the implementation of the VAT reform in 2009, the coefficients of treat * post were not significant within the 95% confidence interval and were all negative. This finding shows that before the implementation of the VAT reform in 2009, domestic companies and foreign-invested companies had the same development trend; specifically, the DID model adopted in this article met parallel trends. In addition, it was found that, after the VAT reform, the total number of patent applications of domestic enterprises was greater than that of foreign-funded enterprises, especially after 2010. The regression coefficients of the model are mostly significantly greater than zero. This shows that the VAT reform has a lagging effect on the innovation of domestic enterprises, and that it did not have a significant impact until 2011. This result not only proves the conclusion of Hypothesis 2, but it is also similar to the conclusion in the existing literature regarding the possible lagging effect of corporate innovation behavior [43].

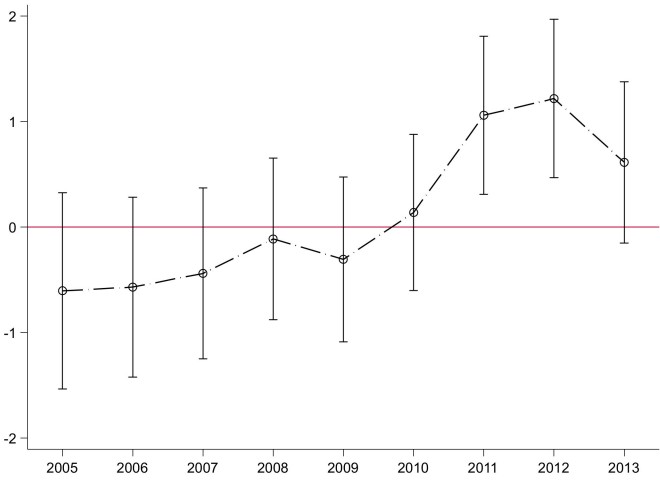

**Figure 4.** Dynamic changes in corporate innovation.

*4.2. Robustness*

(1) Winsor processing was performed on the explained variables to eliminate the influence of outliers on benchmark conclusions. Because the number of patent applications by enterprises varies, the explained variables may have produced outliers, which may have affected the conclusions of our benchmark model. Therefore, we reduced the head and tail of the maximum and minimum values of enterprise innovation performance at the 2.5% level and then used a new sample for regression. The regression results are shown in

the first column of Table 3. It demonstrates that, although the significance has decreased, the coefficient of treat * post remains significantly positive. This result shows that, after we processed the explained variables through Winsor, the conclusions of the benchmark model remain valid.

(2) We removed the interference of the following two policies, which may have interfered with the benchmark conclusion: the interference of the pre-VAT reform policy and the interference of value-added tax changed the business tax policy. The VAT reform first appeared in northeast China in 2004, and was expanded to six provinces in central China in 2007. Therefore, these previous VAT reforms may have also incentivized domestic enterprises in these regions to carry out innovation activities before 2009 and to have better innovation performances after 2009, which will affect the validity of our benchmark conclusions. To solve this problem, we deleted all the data of the northeastern provinces and the data of the six provinces in the central region after 2006 and then used a new sample to perform regression. The regression results are shown in the second column of Table 3. It demonstrates that after excluding the influence of the pre-VAT reform, the coefficient of the treat * post term remains significantly positive. Another policy that needed to be excluded was the pilot reform of the VAT that changed the business tax that began in 2012. Because the reform in 2012 also greatly eased the tax burden of domestic enterprises in the pilot area, were the impact of this policy on corporate innovation not ruled out, it would have also affected the robustness of the conclusions of our benchmark model. To eliminate the impact of this policy, we deleted the sample of enterprises in the pilot areas of the VAT reform after 2011 and re-estimated them. The regression results are shown in the third column of Table 3. It demonstrates that, even after excluding the VAT reform policy, the conclusions of our benchmark model remain robust.

(3) We eliminated the interference of expected effects. Since China's VAT reform is carried out by region and time, when the pilot area implements the VAT reform, it may produce the expected effect of the non-pilot area enterprises and promote the innovative decision-making of the non-pilot area enterprises in advance. In this case, the impact of the 2009 VAT reform on corporate innovation performance may not be exogenous, which will affect the robustness of our benchmark conclusions. To solve this problem, we adopted the idea of Lu and Yu [44] and added the treat * predict item to the benchmark regression model, in which predict is the dummy variable for the 2 years before the impact of the VAT reform policy in 2009, that is, in 2007 and 2008, predict = 1, and in other years, predict = 0. The regression results after adding the treat * predict item are shown in the fourth column of Table 3. It demonstrates that after adding the treat*predict item, although the significance of the treat * post item coefficient has decreased, it remains significant, and treat*, the predict term coefficient, is small and insignificant. This finding shows that the effect of enterprise expectations on enterprise innovation performance is not obvious; thus, the conclusions of the benchmark model in this paper remain stable after controlling for the enterprise's expected effects.

(4) In addition, we still needed to prove the exogenous nature of the 2009 VAT reform policy shock. Drawing on the literature, we adopted the method of counterfactual testing [45]. The year before the 2009 VAT reform was 2008, the year when the policy began to exert its influence. The result of the counterfactual test is shown in the fifth column of Table 3. It demonstrates that the coefficient of the treat * post term is smaller than the coefficient of the benchmark regression, and is insignificant. This finding shows that the 2009 VAT reform policy impact can be regarded as exogenous, and the conclusions of the benchmark model in this paper remain robust.

(5) Finally, to exclude the influence of other unobserved factors that may exist, we drew on the idea of Chetty et al. [46] and adopted the method of randomizing the full sample of companies and producing randomized treatment groups for placebo testing. We randomly selected all the companies in the sample and reassigned these companies as the new treatment group and control group. Next, we used these randomly selected treatment groups and control groups to re-regress the model (1). In order to ensure the robustness

of this kind of random data generation, we used bootstrapping to conduct 500 random sampling inspections. After 500 random sampling tests, results were obtained. The results of the placebo test are shown in Figure 5. Figure 5 shows that the absolute value of the t value of most of the randomly generated treat * post items is less than 2, and for p, the value is greater than 0.1. This finding shows that other unobserved factors have little effect on corporate innovation.

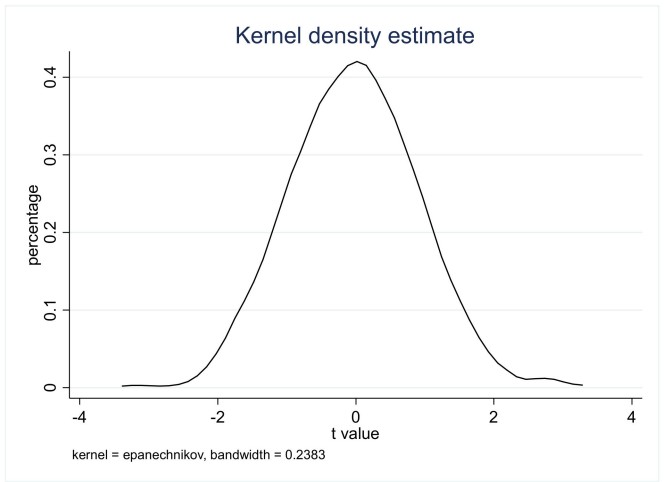

**Figure 5.** Placebo test.

## 5. Further Discussion

### 5.1. Influence Mechanism

In Hypothesis 2, we suggest that the VAT reform promotes corporate innovation through channels that promote corporate investment in fixed assets and eases corporate debt levels. To test whether these two influence channels exist, we used logarithmic corporate fixed asset stock (lnK) and corporate debt ratio (lev) as the proxy variables of fixed asset investment and corporate debt level, respectively. We also used the method of gradually testing regression coefficients pioneered by Baron and Kenny [47] to construct the mediating effect model. The specific method is as follows:

$$y_{icjt} = \beta_1(treat * post) + \beta_c Controls + a_{ct} + a_{jt} + \delta_i + \varepsilon_{it} \tag{3}$$

$$M_{icjt} = \beta_2(treat * post) + \beta_c Controls + a_{ct} + a_{jt} + \delta_i + \varepsilon_{it} \tag{4}$$

$$y_{icjt} = \beta_3(treat * post) + \beta_4 M_{icjt} + \beta_c Controls + a_{ct} + a_{jt} + \delta_i + \varepsilon_{it} \tag{5}$$

where $M_{icjt}$ is an intermediary variable, including logarithmic corporate fixed asset stock and corporate debt ratio. For models (3), (4), and (5), we were most concerned about the coefficients $\beta_2$ and $\beta_4$. If these two coefficients were significant and had the same sign, a mediating effect exists.

The first and second columns of Table 4 report the results of the mediation effect model of the influence of fixed-asset investment channels on enterprise innovation performance. The first column shows that the VAT reform has promoted the increase in the company's logarithmic fixed asset stock. The second column confirms that the logarithmic company's fixed asset stock has a significant positive impact on the company's innovation performance. According to the definition of intermediary effect, we concluded based on the results reported in Table 4 that companies can promote innovation performance through the channel of increasing investment in fixed assets.

**Table 4.** Mediation effect.

|  | (1) | (2) | (3) | (4) |
|---|---|---|---|---|
| Variable | lnK | patent | lev | patent |
| treat * post | 0.051 * | 0.091 *** | −0.014 ** | 0.091 *** |
|  | (1.883) | (2.607) | (−2.063) | (2.609) |
| lnK |  | 0.012 ** |  |  |
|  |  | (2.101) |  |  |
| lev |  |  |  | −0.075 ** |
|  |  |  |  | (−2.091) |
| size | 0.836 *** | −0.083 *** | −0.008 | −0.075 ** |
|  | (23.024) | (−2.754) | (−0.589) | (−2.542) |
| age | 0.067 *** | −0.088 *** | 0.002 | −0.087 *** |
|  | (2.916) | (−3.000) | (0.444) | (−2.989) |
| lique | −0.000 | 0.000 *** | 0.000 *** | 0.000 *** |
|  | (−1.175) | (8.867) | (3.315) | (8.683) |
| SA | −0.126 | 1.097 *** | 0.023 | 1.098 *** |
|  | (−1.272) | (10.831) | (0.773) | (10.907) |
| LBR | 0.035 *** | 0.022 ** | −0.004 *** | 0.022 ** |
|  | (4.012) | (2.353) | (−2.592) | (2.375) |
| Constant | 0.079 | 4.674 *** | 0.670 *** | 4.745 *** |
|  | (0.107) | (3.601) | (2.852) | (3.675) |
| province * year | YES | YES | YES | YES |
| Industry * year | YES | YES | YES | YES |
| N | 102,802 | 102,802 | 103,245 | 103,245 |
| Adj.R$^2$ | 0.158 | 0.069 | 0.044 | 0.069 |

Note: *, **, and *** refer to significance at the 1%, 5%, and 10% levels, respectively.

The third and fourth columns of Table 4 report the results of the mediating effect model of the influence of corporate debt level channels on corporate innovation performance. The first column shows that the VAT reform has significantly alleviated the corporate debt level of enterprises. The second column shows that the corporate debt level has a significant negative impact on corporate innovation. This finding shows that the VAT reform has significantly alleviated the level of corporate debt, and this relief of corporate debt is conducive to the improvement of corporate innovation performance; specifically, companies can promote the improvement of innovation performance through by mitigating corporate debt.

The empirical results shown in Table 4 confirm Hypothesis 2, which indicates that, in the VAT reform, companies have improved corporate innovation performance through the two channels of promoting fixed asset investment and mitigating corporate debt levels.

*5.2. Innovative Pecking Order Effect*

The pecking-order effect implies that, under the condition of financing constraints, companies rank investment projects by investment income to choose the best investment decision [48]. At present, this theory has been applied in the field of trade research [12]. Because of Chinese companies' severe financing constraints (the mean value of the SA index in Table 1 supports this statement) and uncertainty of innovation income, we propose in Hypothesis 3 that Chinese companies will choose the direction of maximizing income to innovate, that is, the direction of explicit gains is observed.

Chinese local governments usually use innovation subsidies to encourage local enterprises to innovate to support the construction of local industries. However, the government's behavior of choosing enterprise innovation instead of the market is often ineffective [49]. Moreover, the government often attaches importance to the quantity of enterprise innovation rather than the quality. Therefore, to obtain explicit short-term benefits such as government subsidies, Chinese companies may choose to innovate in a direction with lower innovation quality in the presence of financing constraints, that is, to conduct strategic innovation [37].

To verify these conjectures and whether Hypothesis 3 is true, we employed the following steps. (1) First, we examined the impact of VAT reform on enterprise innovation choices. Because the VAT reform has eased some of the financing constraints of enterprises, the innovation choices of enterprises in this case often reflect the pecking-order effect of enterprise innovation, that is, the true innovation choices of enterprises. To test the impact of VAT reform on enterprise innovation choices, we separately studied the impact of VAT reform on enterprise invention patent applications and the impact of VAT on enterprise non-invention patent applications. We used this method because of the higher innovation quality of invention patents and the lower innovation quality of non-invention patents [39]. Therefore, analyzing the difference in the impact of the VAT reform on these two types of patent applications can help us understand the true innovation choices of enterprises.

Table 5 shows the empirical evidence of the impact of VAT reform on enterprise invention patent applications and the impact of VAT reform on enterprise non-invention patent applications. The first column is the impact of VAT reform on enterprise invention patent applications. It demonstrates that the VAT reform has not significantly affected enterprise invention patent applications. The second column is the impact of VAT reform on enterprise non-invention patent applications. The difference between the two columns is that the results in the second column show that the VAT reform has a significant impact on the non-invention patent applications of enterprises, which suggests that the impact of the VAT reform on the innovation of domestic enterprises mainly promotes these enterprises to conduct lower-quality innovations.

**Table 5.** Differentiated innovation.

| Variables | (1) Patent 2 | (2) Patent 1 |
|---|---|---|
| treat * post | −0.007 | 0.112 *** |
| | (−0.280) | (3.357) |
| size | −0.135 *** | −0.060 ** |
| | (−6.268) | (−2.121) |
| age | −0.009 | −0.061 ** |
| | (−0.436) | (−2.237) |
| lique | 0.000 *** | 0.000 *** |
| | (8.973) | (3.193) |
| SA | 1.082 *** | 0.867 *** |
| | (13.670) | (8.987) |
| LBR | 0.016 ** | 0.023 *** |
| | (2.409) | (2.646) |
| Constant | 4.921 *** | 3.688 *** |
| | (5.574) | (3.644) |
| province * Year | YES | YES |
| Industry * Year | YES | YES |
| N | 103,254 | 103,254 |
| Adj.R$^2$ | 0.079 | 0.065 |

Note: ** and *** refer to significance at the 5% and 10% levels, respectively.

(2) Although we have confirmed that the VAT reform has promoted enterprises to conduct lower-quality innovations, to further confirm Hypothesis 3, we also needed to prove that this was the pecking-order effect caused by government subsidies. To confirm this view, we needed to test whether the number of innovations of different quality enterprises can have different effects on the enterprises receiving government subsidies. If there was no significant positive relationship between the number of invention patent applications and the company's government subsidies, and the number of non-invention patent applications had a significant positive relationship with the company's government subsidies, government subsidies were considered to have triggered the pecking-order effect of enterprise innovation. To verify this relationship and solve the innovation self-selection problem, we adopted the Heckman two-stage model for testing. For the Heckman

two-stage model, we first created a dummy variable for whether or not the company was innovating and then used company size, company age, SA index, and labor productivity as the covariates for the first stage of regression.

The regression results of the second stage of the Heckman two-stage model are shown in Table 6. After controlling for the province-year fixed effect and the industry-year fixed effect, the inverse Mills ratios (IMR) of the two models were both significant, indicating that the original model has self-selection. The Heckman two-stage model is effective in making corrections. By comparing the relationship between the number of two patent applications and the government subsidies received by enterprises, the results demonstrate that the number of applications for invention patents of enterprises has no significant relationship with the number of government subsidies received by enterprises; however, there was a significant positive relationship between the number of applications for non-invention patents and the enterprises' government subsidies. This finding confirms our conjecture that government subsidies trigger the pecking-order effect of corporate innovation choices, and that this makes the impact of the VAT reform on domestic corporate innovation mainly incentivize these companies to conduct lower-quality innovations; thus, Hypothesis 3 is confirmed.

**Table 6.** The second stage regression result of Heckman test.

| | (1) | (2) |
|---|---|---|
| Variables | Subsidy | Subsidy |
| Patent 2 | −0.062 | |
| | (−1.108) | |
| Patent 1 | | 0.068 ** |
| | | (2.070) |
| size | −0.568 | −0.171 |
| | (−1.201) | (−0.966) |
| age | 0.890 *** | 1.314 *** |
| | (2.755) | (3.415) |
| LBR | −0.068 | 0.052 |
| | (−1.148) | (1.094) |
| SA | 0.156 | −0.080 |
| | (0.235) | (−0.159) |
| SOE | −0.126 | −0.552 * |
| | (−0.292) | (−1.706) |
| IMR | −8.386 * | −13.587 *** |
| | (−1.875) | (−3.228) |
| Constant | 18.886 * | 6.706 |
| | (1.838) | (1.456) |
| province * Year | YES | YES |
| Industry * Year | YES | YES |
| N | 33,319 | 51,369 |
| Adj. $R^2$ | 0.156 | 0.120 |

Note: *, **, and *** refer to significance at the 1%, 5%, and 10% levels, respectively.

### 5.3. Heterogeneity Test

This article analyzed the heterogeneous impact of VAT reform on enterprise innovation performance from two perspectives:

(1) Export heterogeneity. According to the emerging trade theory, export companies tend to have higher productivity [50]; additionally, because of export exposure and learning of cutting-edge technologies in the industry, they will have an export-learning effect [51,52]. All these will promote export companies to have higher innovation performance than non-export companies. Therefore, when the VAT reform reduces the innovation costs of domestic companies, the impact on the innovation performance of export companies should be greater than the impact on the innovation performance of non-export companies. To verify this conjecture, we divided the full sample of companies into exporting companies and non-exporting companies and used the method of model (1) to estimate separately.

The regression results are shown in the first and second columns of Table 7. The results demonstrate that the VAT reform only has a significant and positive impact on export companies and no significant impact on non-export companies.

**Table 7.** Heterogeneity test.

|  | (1) | (2) | (3) | (4) |
|---|---|---|---|---|
| Variable | patent_exp | patent_nonexp | patent_SOE | patent_nonSOE |
| Treat * post | 0.112 ** | −0.083 | −0.115 | 0.096 ** |
|  | (2.326) | (−1.078) | (−0.773) | (2.390) |
| size | −0.036 | −0.081 | −0.218 ** | −0.052 |
|  | (−0.714) | (−1.574) | (−2.197) | (−1.531) |
| age | 0.026 | −0.161 *** | −0.044 | −0.144 *** |
|  | (0.548) | (−3.186) | (−0.637) | (−4.352) |
| lique | 0.000 | 0.000 | 0.000 | 0.000 *** |
|  | (0.122) | (0.221) | (0.143) | (7.663) |
| SA | 1.094 *** | 1.048 *** | 1.294 *** | 1.058 *** |
|  | (6.964) | (5.543) | (4.144) | (8.792) |
| LBR | 0.030 ** | 0.014 | 0.016 | 0.024 ** |
|  | (2.241) | (0.859) | (0.567) | (2.289) |
| Constant | 4.303 *** | 7.310 *** | 5.859 *** | 4.221 *** |
|  | (4.388) | (5.115) | (2.921) | (3.977) |
| province * year | YES | YES | YES | YES |
| Industry * year | YES | YES | YES | YES |
| N | 62,585 | 40,669 | 13,228 | 87,250 |
| Adj.R$^2$ | 0.083 | 0.100 | 0.166 | 0.068 |

Note: ** and *** refer to significance at the 5% and 10% levels, respectively.

(2) Heterogeneity of ownership. China's SOEs tend to have lower financing constraints than private enterprises (non-SOEs; [53]). Therefore, China's state-owned enterprises have a lower debt ratio than non-state-owned enterprises. According to our previous analysis, we believe that the impact of the VAT reform on SOEs should be less than that of non-SOEs. To verify this hypothesis, we divided the full sample of companies into SOEs and non-SOEs and used the method of model (1) to estimate separately. The regression results are shown in the third and fourth columns of Table 7. The results demonstrate that the VAT reform only has a significant and positive impact on non-SOE and has no significant impact on SOE.

## 6. Conclusions

This article used China as an example to study how the tax reforms of the governments of emerging countries affect corporate innovation. Contrary to the literature, we used China's unique tax type, VAT reform, as the research object. We used the combined database of ASIF and the China Innovative Enterprise Database from 2005 to 2013 as the research object to comprehensively examine the impact of China's 2009 VAT reform on Chinese domestic enterprises. Through the DID method, we identified the impact of tax reform on corporate innovation performance and its mechanism. The two main findings were as follows. (1) VAT reform can effectively promote enterprise innovation. This conclusion remained valid after a series of robustness tests. There is a certain time lag in the impact of VAT reform on corporate innovation, and it will have a significant impact after 2011. (2) The VAT reform promotes corporate innovation through channels that promote corporate investment in fixed assets and ease corporate debt ratios. (3) Enterprise innovation also has a pecking-order effect. Although the VAT reform has alleviated some financing constraints of enterprises, driven by the innovation subsidies of the Chinese government, enterprises will choose the innovation strategy with the largest profit when they innovate; specifically, enterprises will pursue the quantity of innovation and ignore the quality of innovation. Therefore, the VAT reform has promoted more low-quality innovations by enterprises. (4) The impact of VAT reform on the innovation performance

of different companies is heterogeneous. Generally, the VAT reform has a greater impact on the innovation performance of export companies and non-SOEs.

On the basis of our research, we propose the following. (1) Emerging countries represented by China should further reduce corporate taxes and fees, continue to deepen tax reforms, and promote the transformation of the industry in the direction of high-quality, sustainable development. (2) The Chinese government should eliminate its subsidy policy guided by industry support, maintain a good market competition environment, and promote market-oriented corporate innovation strategies. (3) The Chinese government should further promote the implementation of tax policies such as export tax rebates, gradually reduce the cost of fixed asset investment by private enterprises, and promote financial reforms to achieve a reasonable and fair financial environment and promote the formation of an effective market competition environment.

**Author Contributions:** Conceptualization, formal analysis, writing—original draft, K.D.; conceptualization, methodology, fund acquisition, H.X.; data curation, supervision, R.Y. All authors have read and agreed to the published version of the manuscript.

**Funding:** This research was funded by the National Social Science Foundation Project of China, grant number 16ZDA038.

**Institutional Review Board Statement:** Not applicable.

**Informed Consent Statement:** Not applicable.

**Data Availability Statement:** Not applicable.

**Acknowledgments:** Authors are thankful to the three anonymous reviewers for their insightful and constructive comments to improve this article.

**Conflicts of Interest:** The authors declare no conflict of interest to disclose.

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
