# Peer review of "Taxation and Enterprise Innovation: Evidence from China’s Value-Added Tax Reform"

_sustainability, doi:10.3390/su13105700_

Round 1

Reviewer 1 Report

In the present paper, the authors assess the impact of taxation on research and development. In order to do that they use a quasi-experiment. Starting in 2009, the Chinese government has changed the VAT rules concerning R&D assets. This new rules apply to Chinese owned companies only. They take the number of patents as a proxy for innovation. They find that those changes had a positive impact on R&D.

I have several questions and remarks

  • The authors should explicitly explain the VAT reform, and they could do that in the introduction.

  • The two first hypothesis should be merged, they are a redundant.

  • Figure 1 and 3 seem to show that nothing is really happening in 2009, and 2012 seems to play a particular role. Figure 2 is more convincing and goes in the same direction as the regressions.

  • Figure 4 is unconvincing, there are only few points and the trends are not obvious.

  • The authors should explain better their “placebo test”. Is it related to bootstrapping? They should add some reference, since it is an interesting point.

Otherwise the paper is well written and could be published.

Author Response

Dear reviewer:

Thank you very much for your valuable evaluation and kind suggestions. We very much appreciate your decision to allow us to further improve our paper. We have studied the reviewers’ comments and suggestions carefully and have made the appropriate revisions. All the revisions are highlighted in blue in the upgraded version. For your and the reviewers’ convenience, we have provided the individual responses below.
We hope that the revised version meets the requirements of the journal.

  • The authors should explicitly explain the VAT reform, and they could do that in the introduction.

  • Response:  Thank you very much for your kind advice. In the updated version, we have revised the introduction according to your suggestion.

  • The two first hypothesis should be merged, they are a redundant.

  • Response: Thank you so much for your constructive comments. And according to your suggestion, I have merged Hypothesis 1 and Hypothesis 2 in 2.2.
  • Figure 1 and 3 seem to show that nothing is really happening in 2009, and 2012 seems to play a particular role. Figure 2 is more convincing and goes in the same direction as the regressions.

  • Response:Thank you for your useful comments. Regarding Figure 1 to Figure 3, we did not describe enough in the original text, so in Section 3.2 we describe these three pictures in detail. And in order to explain why 2009 did not play a particular role but 12 years did a particular role, we added Hypothesis 2 in Section 2.2. By referring to other classic documents, we propose that there may be lag in corporate innovation, and the time trends shown in Figure 1 and Figure 3 are exactly in line with our hypothesis 2;However, we believe that what Figure 2 shows may not be a robust positive relationship, because according to the DID analysis framework, the treatment group and the control group were relatively close in 2009 and 2013, but relatively far from 2010 to 2012.
  • Figure 4 is unconvincing, there are only few points and the trends are not obvious.

  • Response:Thank you for your helpful suggestions. Because we did not explain the results of the event study method in Figure 4 in detail in the original text. According to your proposal, we added a detailed explanation of Figure 4 in section 4.1.
  • The authors should explain better their “placebo test”. Is it related to bootstrapping? They should add some reference, since it is an interesting point.

  • Response:Thank you for your kind suggestion. In section 4.2, we have added reference to the use of placebo test, and also added a detailed introduction to the use of placebo test.

If there are any problems with the paper, please continue to correct me.

yours sincerely

Reviewer 2 Report

Dear authors,

your manuscript has scientific value added. You provided a very consistent and coherent manuscript. My minor recommendations:

  1. The discussion part is crucial part. Typical discussion part is missing in your manuscript  Compare your study to another similiar studies.
  2. Fully depersonalize the manusript, do not use our (delivered, this, provided..) and we (prefer passive voice). 

I hope my comment will be useful for your future work.

Author Response

Dear reviewer:

Thank you for your suggestions on the paper, and we have made the following changes based on your suggestions:
First, we added our discussion on using the DID model in section 3.3 Empirical model based on your suggestion.The red text is the revised draft.
Second, we also removed the personalized names of the text, mostly changed to this article or this paper.
If the paper still has problems. Please point out in time.
yours sincerely

(1) The discussion part is crucial part. Typical discussion part is missing in your manuscript Compare your study to another similar studies.

Response: Thank you very much for your kind advice. I also added a discussion of the use of the DID framework in this article with a red mark in the original text. Similar studies have been listed after the red mark.

(2) Fully depersonalize the manuscript, do not use our (delivered, this, provided..) and we (prefer passive voice).

Response: Thank you very much for your useful suggestions. we also removed the personalized names of the text, mostly changed to this article or this paper.

Reviewer 3 Report

This paper deals with an extremely interesting topic. It is well organized and provides very useful results.

My overall opinion about the study is very positive. I only have a small question: why the evidence ends in 2013? Is it possible to have more recent results? If no, please justify.

Author Response

Dear reviewer:

My overall opinion about the study is very positive. I only have a small question: why the evidence ends in 2013? Is it possible to have more recent results? If no, please justify.  

Response: Thank you very much for your kind advice. But what I need to point out is that in the database we use, whether it is ASIF or The China Innovative Enterprise Database, the data deadline is in 2013. For details, please refer to the website http:/ /microdata.sozdata.com/login.html. So unfortunately we cannot get more recent results, but if there are new updates to these databases in the future, we will further improve our empirical results.

Kind Regards,

Authors

Round 2

Reviewer 3 Report

Taking into account the answer of the authors, I think that the paper can be accepted for publication.